# Selection of Landfill Cover Materials Based on Data Envelopment Analysis (DEA)—A Case Study on Four Typical Covering Materials

**Yibo Zhang** [1] , **Yan Liu** [2], **Xuefeng Min** [2], **Qifan Jiang** [2] **and Weizhou Su** [3,*]

1 School of Emergency Management, Xihua University, Chengdu 610039, China
2 Faculty of Geoscience and Environmental Engineering, Southwest Jiaotong University, Chengdu 611756, China
3 School of Economics and Management, Southwest University of Science and Technology, Mianyang 621010, China
* Correspondence: suweizhou@swust.edu.cn

**Abstract:** Against the background of sustainable development, landfill covers can consist of a range of materials, from clay to geocomposite and polymer composites. Given engineering and environmental requirements, we analyzed the performance and sustainability of four sanitary landfill cover materials, namely clay, HDPE, PVC, and GCL. Within the principles of environmentally sustainable design, we constructed a material selection index based on the performance as well as the economic and environmental impacts of the materials. In addition, using a data envelopment analysis (DEA) model with an analytic hierarchical process (AHP) preference cone, we developed a $C^2WH$ model to evaluate the performance of the selected materials. Through the calculation, we found that the comprehensive indexes of the four covering materials were $E_1 = 0.2600$, $E_2 = 0.5757$, $E_3 = 0.7815$, and $E_4 = 1.0000$, respectively. Our results indicated that the investigated materials could be ranked according to performance as follows: GCL > PVC > HDPE > clay. Thus, our results showed that GCL, with the highest efficiency value, was the optimal cover of the investigated materials. The multiobjective decision model developed in our study can be used as a technical reference and offers support for the selection of eco-friendly landfill cover materials.

**Keywords:** landfill cover material; sustainable design; eco-friendly material selection; DEA model; $C^2WH$ model

## 1. Introduction

During landfill disposal, the decomposition of organic matter in waste can easily lead to the production of landfill gases [1,2]. The gases, primarily composed of methane, are a significant source of greenhouse gas emissions, contributing up to 30% of global anthropogenic greenhouse gas emissions [3]. In addition, odors and other health risks can become problematic in sanitary waste landfills [4]. The selection and use of landfill cover materials are important in reducing landfill-related gas and odor emissions [5]. Specifically, landfill covers can accelerate the conversion of methane into carbon dioxide by promoting the growth of methane-oxidizing bacteria [6,7]. Furthermore, the covering material can effectively filter and prevent the emission of odors and gas, prevent the breeding of mosquitos and flies, and reduce the spread of bacteria [8,9].

Currently, landfill covers are made of either natural materials, which primarily include compacted clay and modified clays with stabilizing additives or synthetic materials, such as geocomposites, polymer composites, and painting materials [10–14]. Natural materials are easily obtainable, affordable, and accompanied by stable engineering performance. However, the high cost of the related construction machinery, which has poor seepage and airproof characteristics, has resulted in natural materials not meeting the requirements of

modern landfills [7,15]. In contrast, artificial materials are ductile, have suitable tensile properties, and are water- and airproof. In addition, as artificial covers are light and thin, the related structures are relatively stable. Under normal circumstances, artificial covers are long-lasting. However, being lightweight, synthetic covers are easily damaged and cannot be reused, leading to a higher economic cost [16]. Therefore, the various cover materials have specific advantages and disadvantages, and the selection of the most appropriate material with the best overall performance is becoming increasingly important. Typically, the selection of a landfill cover material is based on the subjective experience of engineers, from an economic feasibility perspective, or based on the performance of materials, with little consideration of the environmental impacts of the material [17,18].

Within this context and under the guidance of sustainable design, we developed a comprehensive material index system that considers the basic function of various materials, their economic feasibility, and their environmental impact. Thus, using multicriteria decision making (MCDM), we were able to compare and select the most appropriate cover material. In addition, we applied the model using a case study. The findings of our study provide insight into landfill management practices, leading to a reduction in associated landfill gases and, ultimately, increasing the sustainability of landfills.

## 2. Literature Review

The selection of appropriate materials can be complex as it involves multiple evaluation criteria and attribute information, with possible contradictory relationships between attributes, such as economic versus environmental factors [18]. Within this context, MCDM is widely used for the selection of materials. Depending on the specific application, MCDM can be classified into multiple objective decision making (MODM), focusing on objective optimization; and multiple attribute decision making (MADM), focusing on objective comparisons [19]. The decision variables of MODM are continuous and embedded in the region determined by the constraints, resulting in a multiobjective decision-making method with infinite schemes, which are primarily used for optimal design [20,21]. In comparison, MADM has a finite decision scheme, resulting in a discrete multi-objective decision-making method, which primarily focuses on the ranking or preference of finite decision schemes with multiple attributes or indicators [22].

The selection of materials is a typical MADM problem. The classical MADM approach includes a number of material selection methods, such as the analytic hierarchical process (AHP), analytic network process (ANP), vise višekriterijumska optimizacija i kompromisno rešenje (VIKOR), technique for order preference by similarity to ideal solution (TOPSIS), elimination and choice translating reality (ELECTRE), simple additive weighted (SAW), weighted product method (WPM), and data envelopment analysis (DEA) [23–25]. These methods allow materials to be ranked according to specified criteria, and thus offer assistance to decision makers. For example, in the classification and selection of soft and hard magnetic materials, Chauhan and Vaish [26] used the VIKOR method with triangular intuitionistic fuzzy numbers to identify the optimal solution by order of preference. Similarly, Kumar et al. [27] used TOPSIS to evaluate the best material for the design of exhaust pipes, using the cost of materials as an important indicator. In addition, within the context of selecting "sustainable" materials, Zhao [28] used a gray correlation method within MADM for the selection of commercial materials. Using a plastic pipe as an example, the performance of the material was divided into related economic, environmental, and social aspects, with AHP used to assign weights to the performance indicators of alternative materials, while the weighted coefficient of the environmental indicators was increased. Zhou et al. [29] used refrigerator shell material as a case study to investigate the process and mechanical performance, together with the economic and environmental aspects, based on an MODM model integrating neural networks and genetic algorithms. Notably, the selection of eco-friendly materials has not yet been standardized, and there is a need for quantitative research and analyses to guide the related material selection decisions.

In this study, the selection of study materials was based on a combination of function, cost, and environmental impact, with a specific focus on the economic and environmental feasibility of the materials, i.e., the ratio of cost to function or the ratio of environmental impact to cost, which is a typical efficiency-based evaluation. For this task, DEA has stronger adaptability and flexibility.

First proposed by Charnes [30], DEA is now widely used, among others, in business decision making, technology evaluation, and especially in material selection, where it has been successfully applied across a number of different contexts [31,32]. Based on DEA, Dickson [33] used an evaluation index system in the selection of suppliers of construction materials, allowing the suppliers to be ranked according to selected indicators. In addition, Peng et al. [34] developed an improved cross-efficiency DEA model based on entropy weight TOPSIS to solve the optimization problem of wood for furniture. This development allowed for an analysis of the effectiveness and feasibility of the material selection and evaluation model. Furthermore, to improve the economic feasibility of construction material suppliers, Hemmati et al. [35] combined DEA with TOPSIS and used a G-$C^2R$ model in generalized DEA to address various problems, including the reverse order of TOPSIS and the incomplete ordering of DEA. In addition, TOPSIS was used to rank the effective suppliers, thereby providing a basis for decision making. Similarly, Safa et al. [36] combined the AHP and DEA methods and used suppliers of exterior curtain wall engineering as a case study to establish an evaluation index and demonstrate the applicability of the AHP/DEA method in the selection of construction material suppliers.

Because the input and output indexes of the traditional DEA do not contain weighted coefficients, they cannot reflect decision makers' preferences via evaluation indexes, which may lead to a large deviation between theory and practice [37,38]. In particular, this deviation can occur when the DEA model contains a large number of input and output indicators together with unconstrained weights. This results in a disproportionate validity of the decision units and reduces the effectiveness of the model. To solve the constraint of index weight distribution, Charnes and Cooper [39] developed a DEA model with a cone ratio ($C^2WH$ model), which incorporates the preferences of decision makers. On this basis, Wu [40] introduced the concept of the AHP constraint cone by applying the AHP to the relative evaluation of DEA to better reflect the preferences of decision makers. Hahn et al. [41] applied the cone ratio $C^2WH$ model to a technology evaluation index system of road networks and calculated the relative efficiency of each road network system by using the $C^2R$ model and the $C^2WH$ model, separately. Their results indicated that the $C^2WH$ model, which considers the preference of decision makers, was more appropriate than the $C^2R$ model. Thus, the DEA model has unique advantages in analyzing the efficiency of systems with multiple inputs and outputs as well as different dimensions.

In this study, we addressed the issue of material selection using a multiobjective input and output system that considers the function, economic value, and environmental impact of various landfill cover materials. Specifically, we calculated the relative efficiency of each evaluation unit using a DEA-$C^2WH$ model with a preference cone to obtain ranked evaluation values, allowing for the identification of the optimum landfill cover material.

## 3. Materials and Methods

### 3.1. Integrated DEA Approach for Coverage Materials Selection

3.1.1. Evaluation Indicators

In this study, we investigated the function, economic value, and environmental impact of four landfill cover materials and identified sustainability indicators for material selection. From our results, we developed a decision-making matrix with detailed indicators (Table 1) [26,42,43].

**Table 1.** Selected indicators for the selection of landfill cover material.

| Attributes | Indicators | Measurements |
|---|---|---|
| Functional | Permeability coefficient | Obtained from the material manual [44]. |
| | Tensile strength | Obtained from the material manual [44]. |
| | Service life | Obtained from market research. |
| | Landfill compatibility | Combined with on-site investigation and expert scoring, the subjective value assignment method was adopted. The value range is 0–1, with values 0.2, 0.4, 0.6, 0.8, and 1 used to indicate poor, weak, moderate, good, and excellent compatibility, respectively. |
| Economic | Direct cost | $C_p = C_p(x)S$, where $C_p(x)$ is the price per unit mass of cover materials (CNY/m$^2$), and $S$ is the amount of material purchased (m$^2$). |
| | Construction cost | $C_l = \sum_{i=1}^{m} C_l(x)S$, where $m$ is the number of operation processes of material construction, $C_i(x)$ is the construction cost per unit material in the construction operation $i$ (CNY/m$^2$), and $S$ is the construction area of the material (m$^2$). |
| | Usage | The amount of cover material used is determined by the working area of the on-site investigation and is characterized by the thickness of the material. |
| | Recycling cost | $C_R = C_R(x)W$, where $C_R(x)$ represents the recycling cost per unit mass of the cover material (CNY/kg), and $W$ is the recycled mass of the material (kg). |
| Environmental | Energy consumption | Amount of electric energy used per unit of material produced. |
| | Ecological index | The assessment value of the environmental impact, i.e., the ecological index, is obtained from the life cycle assessment method of Eco-indicator 99 (Pt/kg). |
| | Recycling rate | $p_R = \frac{R}{C}$, where $R$ is the amount of recycled material, and $C$ is the material used. |

Material functionality is closely related to the physical and chemical properties of cover materials, including, among others, moisture content, organic matter content, and porosity [16–19]. In this study, we selected the following four function-related indicators: permeability coefficient, tensile strength, service life, and landfill compatibility. Of these, the permeability coefficient of a cover material directly affects the production of landfill gas [45], while the tensile strength reflects the ductility of the cover material in preventing external damage [10]. The service life of a material indicates the lifespan of the material. The longer the service life of the cover material, the less frequently it needs to be replaced, thereby directly affecting landfill efficiency and cost-effectiveness [46]. Landfill compatibility represents the ability of a cover to resist structural changes caused by the material contained in a landfill and its compatibility with the specific landfill environment [47]. Landfill content has a direct impact on the landfill cover, primarily reflected in the temperature adaptability, as well as the corrosion and aging resistance of the material [48,49].

The economic value or aspect of materials primarily focuses on the direct cost of the material, as well as the construction, recycling, and usage costs. Of these, the direct cost is obtained from the market price of a unit of purchased material; the construction cost refers to the cost of placing and installing the landfill cover; the recycling cost refers to the cost of recycling and reprocessing the cover material; and the material usage refers to the amount of the cover material used, which is affected by the thickness of the material, as landfills have a fixed operation area.

The environmental performance of a material primarily relates to the energy consumed, ecological index, and material reuse ratio. Energy consumption refers to the energy consumed during the production process (with electricity consumption as the benchmark), while the ecological index refers to the relative load value of the environmental impact of the material. Using a life cycle assessment, inventory analysis data of selected materials can be categorized according to specific environmental issues. The environmental issues of each category can be normalized according to their related impacts, and the evaluation value

of the environmental impact can be obtained by summation according to the weighted coefficients [50]. The reuse ratio refers to the proportion of the material that can be reused.

### 3.1.2. DEA Model with AHP Constrained Cone

Based on relative efficiency, DEA allows for the evaluation of the relative effectiveness of each decision-making unit (DMU) through multi-indicator input and output analysis. Thus, DEA incorporates the advantages and disadvantages of each alternative, providing an effective way to evaluate multi-objective problems [30,51].

Using $n$ DMUs, each with input and output indicators $X_i$ and $Y_j$, respectively, with the $j_0$th decision-making unit, we used the C$^2$WH model of DEA as follows [51]:

$$s.t. \begin{cases} max\frac{u^T y_0}{v^T x_0}, \\ v^T X - u^T Y \in K, \\ v \in V\{0\}, \\ u \in U \backslash \{0\}, \end{cases} \tag{1}$$

where $X = (x_1, x_2, \cdots x_n)$ is an $m \times n$ matrix; $Y = (y_1, y_2, \cdots y_n)$ is an $s \times s$ matrix; $V \in E_+^m$, $IntV \neq \varnothing$, $U \subset E_+^S$, and $IntU \neq \varnothing$, $K \subset E^n$ are closed convex cones; and $\delta_j = (0, \cdots 0, 1, 0 \cdots, 0)^T \in -K^*(j = 1, 2, \cdots, n)K^* = \{K|K^T K \leq 0, \forall K \in K\}$. $K^*$ is a polar cone of $K$.

Using the Charnes-Cooper linear transformation on the above model, with $t = \frac{1}{v^T x_0}$, $\omega = tv$, and $\mu = tu$, we developed an equivalent model as follows:

$$s.t. \begin{cases} max\mu^T y_0 = V_P, \\ \omega^T X - \mu^T Y \in K, \\ \omega^T x_0 = 1, \\ \omega \in V, \\ \mu \in U. \end{cases} \tag{2}$$

According to the cone duality theory, the model can be expressed as follows:

$$s.t. \begin{cases} min \, \theta = V_D, \\ X\lambda - \theta x_0 \in V^*, \\ -Y\lambda + y_0 \in U^*, \\ \lambda \in -K^*, \end{cases} \tag{3}$$

where $V^*$, $U^*$, and $K^*$ are the polar cones of $U$, $V$, and $K$, respectively; and $V^* = \{v|\hat{v}^T v \leq 0, \forall \hat{v} \in V\}$, $U^* = \{u|\hat{u}^T u \leq 0, \forall \hat{u} \in U\}$, and $K^* = \{k|\hat{k}^T k \leq 0, \forall \hat{k} \in K\}$. Simultaneously, when the effectiveness of the decision unit is measured using a closed convex cone, the production possibility set is calculated as follows:

$$T = \{(x, y)|(x, y) \in (X\lambda, Y\lambda) + (-V^*, U^*), \lambda \in -K^*\} \tag{4}$$

In this study, we used the AHP method to construct the preference cones $V$ and $U$ [52]. Furthermore, we used the C$^2$WH model with a constraint cone based on AHP as follows:

$$s.t. \begin{cases} max\mu^T y_0 = V_P^*, \\ \omega^T X - \mu^T Y \in K, \\ \omega^T x_0 = 1, \\ \omega \in V, \\ \mu \in U. \end{cases} \tag{5}$$

where $V = \{\omega|C\omega \geq 0, \omega \geq 0\}$; $U = \{B\mu \geq 0, \mu \geq 0\}$; and matrices $C$ and $B$ are $m$- and $s$-dimensional matrices, respectively. The model was constructed by establishing a 9-scale judgment matrix, with $\overline{C}_m$ and $\overline{B}_s$ as the input index $X$ and the output index $Y$, respectively.

In addition, a consistency test was conducted on $\overline{C}_m$ and $\overline{B}_m$. After the test, $\lambda_c$ and $\lambda_B$ were set as the maximum eigenvalue of the matrix $\overline{C}_m$ and $\overline{B}_s$, respectively. Thus, $C = \overline{C}_m - \lambda_c E_m$ and $B = \overline{B}_s - \lambda_B E_s$, with $E_m$ and $E_s$ representing the $m$- and $s$-order unit matrices, respectively. This allowed us to form a polyhedral closed convex cone as follows:

$$\begin{cases} Cw \geq 0, & w = (w_1, w_2 \cdots w_m)^T \geq 0, \\ B\mu \geq 0, & \mu = (\mu_1, \mu_2 \cdots \mu_m)^T \geq 0 \end{cases} \tag{6}$$

If the optimal solution $\omega_0, \mu_0$ to a problem $(P)$ satisfies the equation $V_P' = \mu_0^T y_0 = 1$, the decision-making unit $j_0$ is weakly DEA-effective ($C^2WH$). If the optimal solution $\omega_0, \mu_0$ to a problem $(P)$ satisfies the equations $V_P' = \mu_0^T y_0 = 1$ and $\omega_0 \in IntV$ and $\mu \in IntU$, the decision-making unit $j_0$ is DEA-effective ($C^2WH$).

### 3.2. An Illustrative Case Study

In this study, we investigated four landfill cover materials based on market research, namely clay, HDPE geomembrane, PVC geomembrane, and GCL waterproof blanket material (Table 2).

Based on the results of field research on municipal waste landfills in Nanchong City and Chengdu City (Table 1), we obtained the performance index parameters of the four investigated cover materials (Table 3).

**Table 2.** Composition of four investigated landfill cover materials including clay, high-density polyethylene (HDPE) geomembrane, polyvinyl chloride (PVC) geomembrane, and geosynthetic clay liner (GCL) waterproof blanket material.

| Material | Legend | Composition |
|---|---|---|
| Clay |  | Aluminum silicate particles with particle size < 2 μm ($mSiO_2 \cdot n\ Al_2O_3 \cdot xH_2O$) |
| HDPE geomembrane |  | 97.5% HDPE, 2.5% carbon black, antioxidant, and heat stabilizer |
| PVC geomembrane |  | 100% PVC film |
| GCL waterproof blanket |  | A mixture of highly expansive N–bentonite particles, composite geotextiles, nonwoven fabrics, and admixtures |

Within the investigated parameters, the larger the basic performance indicator, the better the cover material. Therefore, basic performance data were used as the output indicator of the evaluation system. On the contrary, the smaller the value of the economic and environmental indicators, the better the cover material. Therefore, the reciprocals of the economic and environmental performance indicators were used as the input indicators

of the evaluation system [53,54]. The specific index values of each of the four investigated materials were further evaluated using DEA after dimensionless treatment according to the percentage (Table 4).

**Table 3.** Index parameters of investigated landfill cover materials, including clay, high-density polyethylene (HDPE) geomembrane, polyvinyl chloride (PVC) geomembrane, and geosynthetic clay liner (GCL) waterproof blanket material.

| Index Parameter | Clay | HDPE | PVC | GCL |
|---|---|---|---|---|
| Permeability coefficient | $1.0 \times 10^{-7}$ | $1.0 \times 10^{-12}$ | $1.2 \times 10^{-12}$ | $1.0 \times 10^{-11}$ |
| Tensile strength (MPa) | 0.015 | 30 | 55 | 15 |
| Service life (years) | 15 | 50 | 50 | 100 |
| Landfill compatibility | 0.6 | 1 | 1 | 0.8 |
| Direct cost (CNY/m$^2$) | 28 | 70 | 70 | 45 |
| Construction cost (CNY/m$^2$) | 9 | 32 | 32 | 16 |
| Usage (mm) | 200 | 1.5 | 1.0 | 15 |
| Recycling cost (CNY/kg) | 5.24 | 2.7 | 4.90 | 5.38 |
| Energy consumption (MJ/kg) | 0.06 | 28.7 | 19.5 | 2.92 |
| Ecological index (millipoint/kg) | 11 | 287 | 170 | 3 |
| Recycling rate (%) | 0.1 | 8.6 | 1.7 | 1.5 |

**Table 4.** Normalization of index parameters of four investigated cover materials, including clay, high-density polyethylene (HDPE) geomembrane, polyvinyl chloride (PVC) geomembrane, and geosynthetic clay liner (GCL) waterproof blanket material.

| | | Index Number | | Clay | HDPE | PVC | GCL |
|---|---|---|---|---|---|---|---|
| Output indicator (*Y*) | Basic performance | 1 | Permeability coefficient | 100 | 0.001 | 0.001 | 0.01 |
| | | 2 | Tensile strength | 0.015 | 30 | 55 | 15 |
| | | 3 | Service life | 15 | 50 | 50 | 100 |
| | | 4 | Landfill compatibility | 0.6 | 1 | 1 | 0.8 |
| Input indicator (*X*) | Economic performance | 5 | Direct cost | 28 | 70 | 70 | 45 |
| | | 6 | Construction cost | 9 | 32 | 32 | 16 |
| | | 7 | Recycling cost | 5.24 | 2.7 | 4.9 | 5.38 |
| | | 8 | Usage | 200 | 1.5 | 1.0 | 15 |
| | Environmental performance | 9 | Energy consumption | 0.06 | 28.7 | 19.5 | 2.92 |
| | | 10 | Ecological index | 11 | 287 | 170 | 3 |
| | | 11 | Recycling rate | 0.1 | 8.6 | 1.7 | 1.5 |

AHP was applied to construct the judgment matrices for the above input and output indicators, separately. The judgment matrices $\overline{C}_m$ and $\overline{B}_s$ were constructed using the 9-scalar method (Table 5) to determine the priority weights of each target.

**Table 5.** The values of the elements in the judgment matrix.

| Value | Statement of Relative Importance |
|---|---|
| 1 | Two elements are equally important |
| 3 | One element is slightly more important than the other |
| 5 | One element is significantly more important than the other |
| 7 | One element is more strongly important than the other |
| 9 | One element is extremely more important than the other |
| 2, 4, 6, 8 | Consider in compromise |

The judgment matrices $\overline{C}_7$ and $\overline{B}_4$ of the input and output indicators were constructed as follows:

$$\overline{C}_7 = \begin{bmatrix} 1 & 1 & 4 & 2 & 6 & 5 & 7 \\ 1 & 1 & 4 & 2 & 6 & 5 & 7 \\ 1/4 & 1/4 & 1 & 1/2 & 3 & 2 & 4 \\ 1/2 & 1/2 & 2 & 1 & 5 & 4 & 6 \\ 1/6 & 1/6 & 1/3 & 1/5 & 1 & 1/2 & 2 \\ 1/5 & 1/5 & 1/2 & 1/4 & 2 & 1 & 3 \\ 1/7 & 1/7 & 1/4 & 1/6 & 1/2 & 1/3 & 1 \end{bmatrix}, \overline{B}_4 = \begin{bmatrix} 1 & 2 & 4 & 5 \\ 1/2 & 1 & 3 & 4 \\ 1/4 & 1/3 & 1 & 2 \\ 1/5 & 1/4 & 1/2 & 1 \end{bmatrix}$$

The above judgment matrix was tested for consistency. For $\overline{C}_7$, where the maximum eigenvalue $\lambda_{max} = 7.1679$, $C \cdot I = \frac{\lambda_{max}}{n-1}$ and $C \cdot R = \frac{C \cdot I}{R \cdot I} = 0.0212 < 0.1$. For $\overline{B}_4$, the maximum eigenvalue $\lambda_{max} = 4.0484$, $C \cdot I = \frac{\lambda_{max}}{n-1} = 0.0161$ and $C \cdot R = \frac{C \cdot I}{R \cdot I} = 0.018 < 0.1$, all of which meet the consistency requirements.

Then, we performed the following transformations:

$$C = \overline{C}_7 - \lambda_C E_m = \begin{bmatrix} -6.1679 & 1 & 4 & 2 & 6 & 5 & 7 \\ 1 & -6.1679 & 4 & 2 & 6 & 5 & 7 \\ 1/4 & 1/4 & -6.1679 & 1/2 & 3 & 2 & 4 \\ 1/2 & 1/2 & 2 & -6.1679 & 5 & 4 & 6 \\ 1/6 & 1/6 & 1/3 & 1/5 & -6.1679 & 1/2 & 2 \\ 1/5 & 1/5 & 1/2 & 1/4 & 2 & -6.1679 & 3 \\ 1/7 & 1/7 & 1/4 & 1/6 & 1/2 & 1/3 & -6.1679 \end{bmatrix}$$

$$B = \overline{B}_4 - \lambda_B E_s = \begin{bmatrix} -3.048 & 2 & 4 & 5 \\ 1/2 & -3.048 & 3 & 4 \\ 1/4 & 1/3 & -3.048 & 2 \\ 1/5 & 1/4 & 1/2 & -3.048 \end{bmatrix}$$

A polyhedral closed convex cone was constructed:

$$\begin{cases} Cw \geq 0, & w = (w_1, w_2 \cdots w_m)^{\mathrm{T}} \geq 0, \\ B\mu \geq 0, & \mu = (\mu_1, \mu_2 \cdots \mu_m)^{\mathrm{T}} \geq 0 \end{cases}$$

## 4. Results and Discussion

Using MATLAB (R2019b, Mathworks, Natick, MA, USA), the data in matrices C and B were entered and run to obtain the input and output data:

$$X_j = \begin{bmatrix} 28 & 70 & 80 & 45 \\ 9 & 32 & 32 & 16 \\ 5.24 & 2.7 & 4.9 & 5.38 \\ 200 & 1.5 & 1.5 & 15 \\ 0.6 & 28.7 & 19.5 & 2.92 \\ 11 & 287 & 170 & 9.6 \\ 10 & 0.116 & 0.588 & 0.667 \end{bmatrix}, Y_j = \begin{bmatrix} 0.00005 & 1 & 1 & 0.1 \\ 0.6 & 1 & 1 & 0.8 \\ 0.015 & 30 & 55 & 15 \\ 15 & 50 & 50 & 100 \end{bmatrix}$$

The C$^2$WH model of the DMU$_1$, i.e., clay covering material, was as follows:

$$\begin{cases} max V_P = 100\mu_1 + 0.015\mu_2 + 15\mu_3 + 0.6\mu_4 \\ 28\omega_1 + 9\omega_2 + 5.24\omega_3 + 200\omega_4 + 0.06\omega_5 + 11\omega_6 + 0.1\omega_7 - 100\mu_1 - 0.015\mu_2 - 15\mu_3 - 0.6\mu_4 \geq 0, \\ 70\omega_1 + 32\omega_2 + 2.7\omega_3 + 1.5\omega_4 + 28.7\omega_5 + 287\omega_6 + 8.6\omega_7 - 0.001\mu_1 - 30\mu_2 - 50\mu_3 - \mu_4 \geq 0, \\ 70\omega_1 + 32\omega_2 + 4.9\omega_3 + 1.0\omega_4 + 19.5\omega_5 + 170\omega_6 + 1.7\omega_7 - 0.001\mu_1 - 55\mu_2 - 50\mu_3 - \mu_4 \geq 0, \\ 45\omega_1 + 16\omega_2 + 5.38\omega_3 + 15\omega_4 + 2.92\omega_5 + 3\omega_6 + 1.5\omega_7 - 0.01\mu_1 - 15\mu_2 - 100\mu_3 - 0.8\mu_4 \geq 0, \\ 28\omega_1 + 9\omega_2 + 5.24\omega_3 + 200\omega_4 + 0.06\omega_5 + 11\omega_6 + 0.1\omega_7 = 1, \\ CW \geq 0, BU \geq 0, \\ W \geq 0, U \geq 0 \end{cases}$$

Similarly, the C$^2$WH model of DMU$_2$, i.e., HDPE geomembrane, was as follows:

$$
\begin{cases}
maxV_P = 0.001\mu_1 + 30\mu_2 + 50\mu_3 + \mu_4 \\
28\omega_1 + 9\omega_2 + 5.24\omega_3 + 200\omega_4 + 0.06\omega_5 + 11\omega_6 + 0.1\omega_7 - 100\mu_1 - 0.015\mu_2 - 15\mu_3 - 0.6\mu_4 \geq 0, \\
70\omega_1 + 32\omega_2 + 2.7\omega_3 + 1.5\omega_4 + 28.7\omega_5 + 287\omega_6 + 8.6\omega_7 - 0.001\mu_1 - 30\mu_2 - 50\mu_3 - \mu_4 \geq 0, \\
70\omega_1 + 32\omega_2 + 2.7\omega_3 + 1.5\omega_4 + 28.7\omega_5 + 287\omega_6 + 8.6\omega_7 - 0.001\mu_1 - 30\mu_2 - 50\mu_3 - \mu_4 \geq 0, \\
45\omega_1 + 16\omega_2 + 5.38\omega_3 + 15\omega_4 + 2.92\omega_5 + 3\omega_6 + 1.5\omega_7 - 0.01\mu_1 - 15\mu_2 - 100\mu_3 - 0.8\mu_4 \geq 0, \\
70\omega_1 + 32\omega_2 + 2.7\omega_3 + 1.5\omega_4 + 28.7\omega_5 + 287\omega_6 + 8.6\omega_7 = 1, \\
CW \geq 0, BU \geq 0, \\
W \geq 0, U \geq 0
\end{cases}
$$

The C$^2$WH model of DMU$_3$, i.e., PVC covering material, was as follows:

$$
\begin{cases}
maxV_P = 0.001\mu_1 + 55\mu_2 + 50\mu_3 + \mu_4 \\
28\omega_1 + 9\omega_2 + 5.24\omega_3 + 200\omega_4 + 0.06\omega_5 + 11\omega_6 + 0.1\omega_7 - 100\mu_1 - 0.015\mu_2 - 15\mu_3 - 0.6\mu_4 \geq 0, \\
70\omega_1 + 32\omega_2 + 2.7\omega_3 + 1.5\omega_4 + 28.7\omega_5 + 287\omega_6 + 8.6\omega_7 - 0.001\mu_1 - 30\mu_2 - 50\mu_3 - \mu_4 \geq 0, \\
70\omega_1 + 32\omega_2 + 4.9\omega_3 + 1.0\omega_4 + 19.5\omega_5 + 170\omega_6 + 1.7\omega_7 - 0.001\mu_1 - 55\mu_2 - 50\mu_3 - \mu_4 \geq 0 \\
45\omega_1 + 16\omega_2 + 5.38\omega_3 + 15\omega_4 + 2.92\omega_5 + 3\omega_6 + 1.5\omega_7 - 0.01\mu_1 - 15\mu_2 - 100\mu_3 - 0.8\mu_4 \geq 0, \\
70\omega_1 + 32\omega_2 + 4.9\omega_3 + 1.0\omega_4 + 19.5\omega_5 + 170\omega_6 + 1.7\omega_7 = 1, \\
CW \geq 0, BU \geq, \\
W \geq 0, U \geq 0
\end{cases}
$$

The C$^2$WH model of DMU$_4$, i.e., GCL waterproof blanket, was as follows:

$$
\begin{cases}
maxV_P = 0.01\mu_1 + 15\mu_2 + 100\mu_3 + 0.8\mu_4 \\
28\omega_1 + 9\omega_2 + 5.24\omega_3 + 200\omega_4 + 0.06\omega_5 + 11\omega_6 + 0.1\omega_7 - 100\mu_1 - 0.015\mu_2 - 15\mu_3 - 0.6\mu_4 \geq 0, \\
70\omega_1 + 32\omega_2 + 2.7\omega_3 + 1.5\omega_4 + 28.7\omega_5 + 287\omega_6 + 8.6\omega_7 - 0.001\mu_1 - 30\mu_2 - 50\mu_3 - \mu_4 \geq 0, \\
70\omega_1 + 32\omega_2 + 4.9\omega_3 + 1.0\omega_4 + 19.5\omega_5 + 170\omega_6 + 1.7\omega_7 - 0.001\mu_1 - 55\mu_2 - 50\mu_3 - \mu_4 \geq 0, \\
45\omega_1 + 16\omega_2 + 5.38\omega_3 + 15\omega_4 + 2.92\omega_5 + 3\omega_6 + 1.5\omega_7 - 0.01\mu_1 - 15\mu_2 - 100\mu_3 - 0.8\mu_4 \geq 0, \\
45\omega_1 + 16\omega_2 + 5.38\omega_3 + 15\omega_4 + 2.92\omega_5 + 3\omega_6 + 1.5\omega_7 = 1, \\
CW \geq 0, BU \geq 0, \\
\geq 0, U \geq 0
\end{cases}
$$

MATLAB software was run to solve the above four planning equations. See Table 6 for the output results. Using the developed C$^2$WH model, we evaluated the efficiency (E) of the four investigated materials. The outputs of the model were E1 = 0.2600, E2 = 0.5757, E3 = 0.7815, and E4 = 1.0000. Our results indicated that DMU$_4$ was relatively effective among the four materials, with GCL having the highest efficiency evaluation value. In terms of performance, the four investigated materials could be ranked as follows: GCL > PVC > HDPE > clay.

As GCL obtained the highest efficiency evaluation value, it could be interpreted as being the optimal landfill cover material. Specifically, GCL waterproof blanket material prevents seepage, occupies less space, is easy to install, and is cost-effective. In addition, GCL waterproof blanket material is self-repairing and can increase the self-waterproofing effect of the cover [55,56]. These characteristics explain the recent trend of replacing or partially replacing clay cover materials with GCL waterproof blankets. In terms of sustainable design, GCL waterproofing blankets have lesser environmental and social impacts than other materials. This result is consistent with the results of other studies on the mechanics, permeability, and stability of long-term GCL landfill covers [57,58]. This agreement indicates that the C$^2$WH model used in our study can be applied to further research on landfill cover materials.

PVC materials and HDPE geomembranes are both composed of artificial composite materials. Notably, PVC has slightly better environmental properties, a less energy-intensive production process, and less of an environmental impact related to the production life cycle than HDPE geomembranes, resulting in better overall performance. HDPE geomembranes and clay covers are currently the most commonly used cover materials for landfills. This highlights the discrepancy between the results of our model and commonly used engineering practices. Furthermore, this discrepancy between theory and practice underlines the shortcomings of material selection through traditional empirical methods, resulting in a greater impact on the environment. Notably, based on sustainability theory, HDPE geomembranes and clay materials, which have a high energy consumption, high ecological index, and low recycling rate, are not optimal cover materials and should gradually be replaced by materials with better performance [59,60]. Within this context, we suggest that GCL waterproof

blankets, composed of a variety of materials, should be used for landfill covers. GCL covers prevent seepage, are easy to install, and can decrease the environmental problems caused by the antiseepage defects of monocomponent cover materials.

**Table 6.** Evaluation results of four investigated cover materials including clay, high density polyethylene (HDPE) geomembrane, polyvinyl chloride (PVC) geomembrane, and geosynthetic clay liner (GCL) waterproof blanket material.

| Cover Material | Clay | HDPE | PVC | GCL |
|---|---|---|---|---|
| $E$ | 0.2600 | 0.5757 | 0.7815 | 1.0000 |
| | 0.0325 | 0.0140 | 0.0105 | 0.0152 |
| | 0.0000 | 0.0000 | 0.0000 | 0.0087 |
| | 0.0117 | 0.0000 | 0.0000 | 0.0092 |
| $W$ | 0.0000 | 0.0018 | 0.0032 | 0.0064 |
| | 0.0017 | 0.0006 | 0.0006 | 0.0018 |
| | 0.0026 | 0.0000 | 0.0009 | 0.0023 |
| | 0.0000 | 0.0004 | 0.0004 | 0.0013 |
| | 0.0189 | 0.0128 | 0.0214 | 0.0289 |
| $\mu$ | 0.0565 | 0.0260 | 0.0272 | 0.0293 |
| | 0.0000 | 0.0097 | 0.0101 | 0.0009 |
| | 0.0151 | 0.0049 | 0.0035 | 0.0096 |

## 5. Conclusions and Further Study

Combined with the market research on the current situation of sanitary landfill cover material use, in this study, we evaluated the performance, economic value, and environmental properties of selected materials using a comprehensive analysis that combined qualitative and quantitative aspects. The $C^2WH$ model with preference cone was used to establish the evaluation model of cover material to achieve the optimal selection of sanitary landfill cover material.

Based on the concept of sustainable design, four typical landfill cover materials were investigated, namely clay, HDPE, PVC, and GCL, using a combined AHP and DEA material selection and evaluation method. Specifically, we developed an evaluation index system, with economic and environmental performance representing the input index and the basic performance of material as the output index. The $C^2WH$ model with a preference cone was successfully used to evaluate and rank the investigated landfill cover materials, with results indicating that GCL material was the optimal cover material. This result provides a reference and data support for multiobjective decision-making problems regarding material selection.

Although we included a number of factors in the developed evaluation index system, some influencing factors, such as landfill gas generated in the landfill process, fell beyond the scope of this study. In addition, we used the $C^2WH$ model with preference cones to reflect the subjective judgment of decision makers. In light of clear individual differences in subjective preferences, which can easily influence the assignment of weights and evaluation results, further research is needed regarding sensitivity analyses. In the next step, we will construct a more complete index system and select representative performance indicators from a broader range. In addition to a single cover layer material, according to the specific engineering application, a composite liner system can be considered, such as HDPE + GCL or clay + HDPE geomembrane and other composite cover structures, and then more comprehensively examine the material performance of sanitary landfill cover layer.

**Author Contributions:** Conceptualization, W.S. and Y.Z.; methodology, Y.Z. and Y.L.; software, Q.J.; validation, X.M.; formal analysis, Y.L., X.M. and Q.J.; data curation, W.S.; writing—original draft preparation, Y.Z.; writing—review and editing, W.S. All authors have read and agreed to the published version of the manuscript.

**Funding:** Funding was received from the Sichuan Youth Science and Technology Innovation Team Funding (2022JDTD0005), Fundamental Research Funds for the Central Universities (2682021ZTPY088), and Natural Science Foundation of Sichuan, China (2022NSFSC0240).

**Institutional Review Board Statement:** Not applicable.

**Informed Consent Statement:** Not applicable.

**Data Availability Statement:** The data are available upon reasonable request from the corresponding author.

**Conflicts of Interest:** The authors declare no conflict of interest.

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
