# Peer review of "Selection of Landfill Cover Materials Based on Data Envelopment Analysis (DEA)—A Case Study on Four Typical Covering Materials"

_sustainability, doi:10.3390/su141710888_

Round 1

Reviewer 1 Report

The manuscript title “Selection of landfill cover materials based on data envelopment analysis (DEA)” presents an interesting study and should be published after major revision.

1.       The structure of the work is disturbed.  According to Sustainability’s Instruction for Authors (https://www.mdpi.com/journal/sustainability/instructions) the manuscript should be organized according to IMRAD scheme. In the paper, there is no clearly separated description of the research methodology.  Scarce information on the research methodology appears only in Chapter 4. The descriptions of the next stages of research are missing. In my opinion, the description of the research methodology should be included before “Evaluation indicators”.

2.       In the introduction, please expand the characteristic of materials used as landfill covers. Only clay, or synthetic materials such as geocomposites, polymer composites, and painting materials  were mentioned as the materials used. Nowadays expansive clays with stabilizing additives are also very often used, which significantly improve the properties of the natural covers. Additionally, the use of waste materials (e.g. zeolites) as additives is part of sustainable development (e.g . 1) Widomski, M.K.; Musz-Pomorska, A.; Franus, W. Hydraulic and Swell–Shrink Characteristics of Clay and Recycled Zeolite Mixtures for Liner Construction in Sustainable Waste Landfill. Sustainability 2021, 13, 7301. https://doi.org/10.3390/su13137301;  2) Widomski M.K., Stepniewski W., Horn R., Bieganowski A, Gazda L, Franus M., PawÅ‚owska M., Shrink-swell potential, hydraulic conductivity and geotechnical properties of clay materials for landfill liner construction. 2015. Int. Agrophys., nr 3, vol. 29, s. 365-375; 3) M.K. Widomski, W. StÄ™pniewski, A. Musz-Pomorska, Clays of Different Plasticity as Materials for Landfill Liners in Rural Systems of Sustainable Waste Management, Sustainability 2018, 10(7), 2489; https://doi.org/10.3390/su10072489)

3.       Please specify, the meaning of symbols in the table 5 (E, W, μ) and how they were calculated.

4.       There are no general and specific conclusions in the paper. The results must be interpretive rather than just descriptive and the research results should be related to the relevant literature citations for validity and reliability.

Reviewer 2 Report

The article is quite interesting, but due to its severely limited extent (and brief form) it should be treated as a case study. Therefore, my first suggestion is to expand the title of the article by adding " - case study" after the current title. The rationale for such a proposal is that the authors have added little new knowledge to the science and limited themselves to a single analysis, and could have treated the topic a little more broadly. The authors wrote that they took into account the preferences of decision makers. This aspect, meanwhile, could have been expanded and there could have been more investigation into how these preferences affect outcomes. For example, could a change in preferences lead to an outcome in which the objectively worst material gets the best performance indicator? And if so, how can such a scenario be avoided or made less likely. Such an analysis is not difficult to perform, it is sufficient to assume that the subjective preferences are variable and taken variantly in subsequent calculations.

An additional drawback of the article is the very brief treatment of the method itself, which the authors write they have developed. It is described in section 3.2 in an extremely short and hermetic manner. Many of the symbols presented there are not explained, nor is the mere 'mechanism' of the method explained. It should be assumed that not all readers of Sustainability magazine are experts in linear-topological spaces, so a more extended and accessible description of the method used is advisable. As is an explanation of what is hidden in Table 5 under the symbols E, W and miu.

Another remark concerns the not entirely correct development of the abbreviation of the VIKOR method, which should be "višekriterijumska optimizacija i kompromisno rešenje" (letters "V" and "I" are both from the first word). Besides, it is worth stating that this is the name in Serbian, which in English means "multicriteria optimisation and compromise solution".

There are also at least two inconsistencies to be found in the text. In Table 1, one of the indicators is called 'Landfill adaptability', and later in Tables 3 and 4 it appears as 'Landfill compatibility'. This introduces unnecessary confusion into the content. In the second case, the authors first write in lines 232-237 that the environmental indicators were used as input indicators and then in the conclusion that they were used as output indicators. It is strange that the authors, when writing the text of the article, did not remember the rather fundamental assumption on which they based their calculations.

Furthermore, on lines 96-98 the authors write that the methods described above only take into account performance and economic aspects of materials. Meanwhile, only a few lines above, the authors, when discussing two items of literature, indicate that environmental aspects were taken into account. They are present in, the methods for assessing materials, and even constitute an indispensable element of them, as for example in the method described in https://doi.org/10.21307/acee-2019-009. The authors should therefore not emphasise so much the absence of this aspect in the analyses for optimal material selection. Although I agree that there is undoubtedly still much room for further research and analysis to improve these methods.

In summary, I believe that the article requires a major revision before it is submitted for publication. 

Round 2

Reviewer 1 Report

The manuscript has been sufficiently improved.

Author Response

We appreciate the reviewer for this remark.

Reviewer 2 Report

I am satisfied with the changes made.

Author Response

(The authors gave the same response as above.)
